# Negative Preference Optimization: From Catastrophic Collapse to Effective Unlearning

**Ruiqi Zhang & Licong Lin** [*]
Department of Statistics, UC Berkeley
{rqzhang, liconglin}@berkeley.edu

**Yu Bai**
Salesforce AI Research
yu.bai@salesforce.com

**Song Mei**
Department of Statistics, UC Berkeley
songmei@berkeley.edu

## Abstract

Large Language Models (LLMs) often memorize sensitive, private, or copyrighted data during pre-training. LLM unlearning aims to eliminate the influence of undesirable data from the pre-trained model while preserving the model's utilities on other tasks. Several practical methods have recently been proposed for LLM unlearning, mostly based on gradient ascent (GA) on the loss of undesirable data. However, on certain unlearning tasks, these methods either fail to effectively unlearn the target data or suffer from catastrophic collapse—a drastic degradation of the model's utilities.

In this paper, we propose *Negative Preference Optimization* (NPO), a simple alignment-inspired method that could efficiently and effectively unlearn a target dataset. We theoretically show that the progression toward catastrophic collapse by minimizing the NPO loss is exponentially slower than GA. Through experiments on synthetic data and the benchmark TOFU dataset, we demonstrate that NPO-based methods achieve a better balance between unlearning the undesirable data and maintaining the model's utilities. We also observe that NPO-based methods generate more sensible outputs than GA-based methods, whose outputs are often gibberish. Remarkably, on TOFU, NPO-based methods are the first to achieve reasonable unlearning results in forgetting 50% (or more) of the training data, whereas existing methods already struggle with forgetting 10% of training data.

## 1 Introduction

Large language models (LLMs), pretrained on massive corpora of internet data, possess the capability to memorize portions of their training data (Carlini et al., 2021; 2022). However, this capability raises significant concerns, as the training data may contain sensitive or private information, potentially leading to societal challenges. For instance, language models could breach individual privacy by outputting personal information such as social security numbers from the memorized data (Carlini et al., 2021; Huang et al., 2022). They might also violate copyright by generating text from memorized books, such as the Harry Potter novels (Eldan & Russinovich, 2023). Furthermore, LLM assistants for biology could inadvertently aid in the development of biological weapons by troubleshooting bottlenecks, increasing the risk of such attempts (Sandbrink, 2023; Li et al., 2024). In response to these concerns, regulations like the EU's General Data Protection Regulation (GDPR) (Mantelero, 2013; Voigt & Von dem Bussche, 2017) and the US's California Consumer Privacy Act (CCPA) (CCPA, 2018) have mandated *the Right to be Forgotten*, requiring applications to support the deletion of information contained in training samples upon user requests. This has motivated a line of research on *machine unlearning*, aiming to address these challenges.

---

[*]Equal contributions; the more junior author is listed earlier.
Code is available at: https://github.com/licong-lin/negative-preference-optimization.

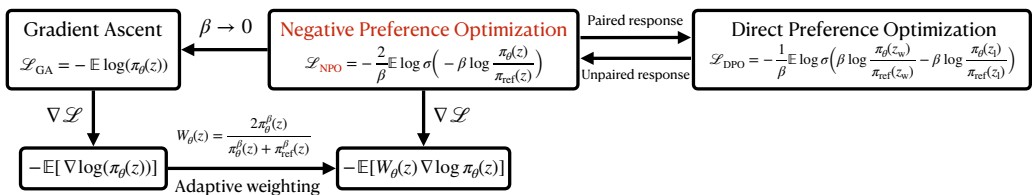

**Figure 1:** Gradient Ascent (GA), Negative Preference Optimization (NPO), and Direct Preference Optimization (DPO). NPO can be interpreted as DPO without positive samples. The gradient of NPO is an adaptive weighting of that of GA, and the weight vanishes for unlearned samples.

Machine unlearning (Cao & Yang, 2015; Bourtoule et al., 2021) aims to delete the influence of specific training samples from machine-learning models while preserving other knowledge and capabilities (Liu et al., 2024a; Zhang et al., 2023; Nguyen et al., 2022; Xu et al., 2023; Si et al., 2023). Notably, a straightforward approach to unlearning is to retrain a language model from scratch. However, as retraining from scratch is typically computationally expensive, cheaper methods for removing undesirable information is highly desirable. Recently, several works (Jang et al., 2022; Wang et al., 2023; Chen & Yang, 2023; Yao et al., 2023; Eldan & Russinovich, 2023; Yao et al., 2024; Liu et al., 2024b; Li et al., 2024) proposed scalable and practical techniques for unlearning LLMs through directly fine-tuning the trained model. Core to many of these works is a *gradient ascent* procedure on the prediction loss over the dataset to be unlearned (the *forget set*), building on the intuition that gradient ascent is an approximation of "reverting" gradient descent optimization on the forget set.

Despite its simplicity and widespread use, the performance of gradient ascent based approaches remain unsatisfactory. A notable example concerns the recently released benchmark dataset TOFU (Maini et al., 2024), which consists of synthetically generated biographies of 200 fictitious authors, and the task is to unlearn the biographies of 1%, 5%, and 10% of the 200 authors from a model that is already fine-tuned on all 200 authors. In their evaluation of forgetting 10% of the authors, Maini et al. (2024) demonstrated that gradient ascent and its variants fail to provide a satisfactory balance between forget quality (the difference between the unlearned model and retrained model evaluated on the forget set) and model utility (the general performance on other tasks).

In this work, we begin by observing that gradient ascent can often cause a rapid deterioration of model utility during training—a phenomenon we term *catastrophic collapse*—which we believe is responsible for its unsatisfactory performance. Towards fixing this, we propose a simple yet effective objective function for unlearning termed *Negative Preference Optimization* (NPO). NPO takes inspiration from preference optimization (Rafailov et al., 2024; Ouyang et al., 2022; Bai et al., 2022), and can be viewed as a variant of preference optimization that only uses negative samples. Through both theory and experiments, we show that NPO resolves the catastrophic collapse issue associated with gradient ascent, provides more stable training dynamics, and achieves a better trade-off between forgetting quality and model utility. Coupled with a cross-entropy loss on the retain set, NPO achieves state-of-the-art performance on the TOFU dataset, and achieves the first non-trivial unlearning result on the challenging task of forgetting 50% of the TOFU data.

**Summary of contributions and paper outline.**

- We outline existing gradient ascent based methods for machine unlearning, and find that these methods suffer from *catastrophic collapse* (Section 2). We identify the linear divergence speed of gradient ascent as a main reason for catastrophic collapse.

- We introduce Negative Preference Optimization (NPO), a simple alignment-inspired loss function for LLM unlearning that addresses the catastrophic collapse issue of gradient ascent (GA; Section 3). We demonstrate that NPO reduces to gradient ascent (GA) in the high-temperature limit. We show in theory the progression towards catastrophic collapse when minimizing the NPO loss is exponentially slower than with GA. See Figure 1 for an illustration of NPO and its connections with existing objectives.

- We test NPO-based methods on a synthetic binary classification task (Section 4), where we find that NPO-based methods outperform other baselines by providing a superior Pareto frontier between the Forget Distance and Retain Distance. Furthermore, NPO-based methods exhibit greater learning stability compared to GA-based methods.

- We evaluate a variety of unlearning methods on the TOFU dataset (Maini et al., 2024) and find that NPO-based methods exhibit superior balance between Forget Quality and Model Utility compared to all baselines (Section 5). Additionally, NPO-based methods improve the stability of the unlearning process and the readability of the output. Notably, we show that NPO-based methods are the only effective unlearning methods for forgetting 50%-90% of the data, a significant advance over all existing methods which already struggle with forgetting 10% of the data (Section 5.3).

There is a vast literature on machine unlearning and LLM unlearning. Due to limited space, we discuss these related work in Appendix A.

## 2 Preliminaries on Machine Unlearning

**Machine Unlearning** refers to the following problem: Given an *initial model* (also the *reference model*) $\pi_{\mathrm{ref}}(y|x)$ that is already trained on a dataset $\mathcal{D} = \{(x_i, y_i)\}_{i \in [n]}$, how to make the model *forget* a specific subset (henceforth the *forget set*) $\mathcal{D}_{\mathrm{FG}} \subseteq \mathcal{D}$ of the training data? More precisely, we aim to fine-tune[1] the model to make it behave like the *retrained model* $\pi_{\mathrm{retr}}$, a model trained only on the *retain set* $\mathcal{D}_{\mathrm{RT}} = \mathcal{D} \setminus \mathcal{D}_{\mathrm{FG}}$. In other words, we would like the model to behave as if the samples in the forget set $\mathcal{D}_{\mathrm{FG}}$ were never used to train it.

By definition, the best approach for machine unlearning in principle is to retrain the model from scratch on $\mathcal{D}_{\mathrm{RT}}$ only which is however often intractable in practice.

**Gradient ascent** is a key component in many existing LLM unlearning methods, and also an important baseline method for LLM unlearning on its own. The idea is simply to perform gradient ascent on the (next-token prediction) loss over the forget set, which can be viewed equivalently as gradient descent on the *negative* prediction loss, denoted as $\mathcal{L}_{\mathrm{GA}}$:

$$\mathcal{L}_{\mathrm{GA}}(\theta) = -\underbrace{\mathbb{E}_{\mathcal{D}_{\mathrm{FG}}}[-\log(\pi_\theta(y|x))]}_{\text{prediction loss}} = \mathbb{E}_{\mathcal{D}_{\mathrm{FG}}}[\log(\pi_\theta(y|x))]. \tag{1}$$

The rationale of gradient ascent is that since the initial model $\pi_{\mathrm{ref}}$ is trained on $\mathcal{D} = \mathcal{D}_{\mathrm{FG}} \cup \mathcal{D}_{\mathrm{RT}}$, a subsequent *maximization* of prediction loss on the forget set $\mathcal{D}_{\mathrm{FG}}$ would approximately "revert" the optimization on the forget set $\mathcal{D}_{\mathrm{FG}}$, thus unlearning $\mathcal{D}_{\mathrm{FG}}$ and approximating a model trained on $\mathcal{D}_{\mathrm{RT}}$ only.

**Other loss functions.** Building on gradient ascent, a large class of unlearning methods perform gradient-based optimization on a linear combination of the GA loss $\mathcal{L}_{\mathrm{GA}}$ and several other loss functions that encourage unlearning (Jang et al., 2022; Yao et al., 2023; Chen & Yang, 2023; Maini et al., 2024; Eldan & Russinovich, 2023). Notable examples include

- Forget (FG) loss: $\mathcal{L}_{\mathrm{FG}}(\theta) = -\mathbb{E}_{\mathcal{D}_{\mathrm{FG}}}[\log(\pi_\theta(\tilde{y}|x))]$, where $(x, y) \sim \mathcal{D}_{\mathrm{FG}}$ and $\tilde{y} \neq y$ is any "uninformed" response for prompt $x$ which the unlearned model could aim to output. Examples of such $\tilde{y}$'s include replacing true information by random (but appearingly sensible) information (which requires hand-crafting such as Eldan & Russinovich (2023)), or simply answering "I don't know" (Maini et al., 2024).

- Retain (RT) loss: $\mathcal{L}_{\mathrm{RT}}(\theta) = -\mathbb{E}_{\mathcal{D}_{\mathrm{RT}}}[\log(\pi_\theta(y|x))]$, which encourages the model to still perform well on the retain set $\mathcal{D}_{\mathrm{RT}}$;

- $\mathcal{K}_{\mathrm{FG}}(\theta) = \mathbb{E}_{\mathcal{D}_{\mathrm{FG}}}[\mathrm{D}(\pi_\theta(\cdot|x)||\pi_{\mathrm{ref}}(\cdot|x))]$, which measures the distance to the initial model $\pi_{\mathrm{ref}}$ (in KL divergence) on the forget set;

- $\mathcal{K}_{\mathrm{RT}}(\theta) = \mathbb{E}_{\mathcal{D}_{\mathrm{RT}}}[\mathrm{D}(\pi_\theta(\cdot|x)||\pi_{\mathrm{ref}}(\cdot|x))]$, which measures the distance to the initial model $\pi_{\mathrm{ref}}$ (in KL divergence) on the retain set.

---

[1]There are alternative approaches such as prompt engineering (Pawelczyk et al., 2023) for performing unlearning tasks.

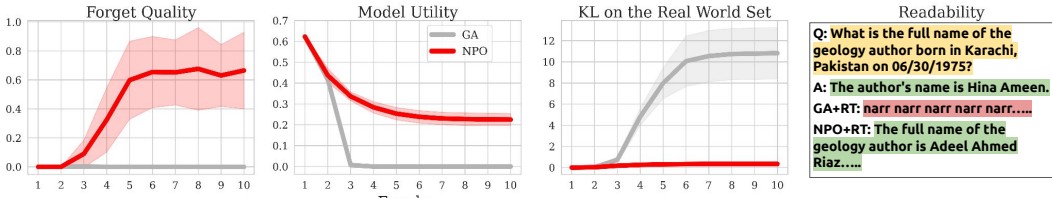

**Figure 2:** Comparison between GA and NPO on forget quality, model utility, KL divergence on the real-world Set, and the answers to the forget set. The rightmost figure shows the answers generated from variants of GA and NPO that incorporates the RT loss. All figures are generated on the Forget05 task in the TOFU data, trained for 10 epochs (detailed setup in Appendix E.1).

For example, Yao et al. (2023) minimize a combination of $\{\mathcal{L}_{\mathrm{GA}}, \mathcal{L}_{\mathrm{FG}}, \mathcal{K}_{\mathrm{RT}}\}$, and Chen & Yang (2023) minimize a combination of $\{\mathcal{L}_{\mathrm{GA}}, \mathcal{L}_{\mathrm{RT}}, -\mathcal{K}_{\mathrm{FG}}, \mathcal{K}_{\mathrm{RT}}\}$. Maini et al. (2024) find that incorporating the retain loss $\mathcal{L}_{\mathrm{RT}}$ improves the performance on various unlearning methods.

**Forget quality and model utility.** Unlearning methods should not only unlearn the forget set, i.e. achieve a high *forget quality*, but also maintain the model's performance on the retain set, i.e. maintain the *model utility*. For example, letting the model to simply output "I don't know" is a unlearning method that achieves good forget quality (in certain sense) but bad model utility. While there is not yet a consensus on the right metrics for forget quality and model utility (and we will present our choices momentarily), a general rule of thumb is that unlearning methods should achieve a good tradeoff between these two goals.

## 2.1 Catastrophic collapse of gradient ascent

We begin by testing gradient ascent as a standalone method (as opposed to combining it with other losses), and find that gradient ascent exhibits a common failure mode dubbed as *catastrophic collapse*: Along the unlearning process, the model utility quickly drops to zero, and the forget quality improves temporarily for a very short time horizon before quickly dropping too (Figure 2 left/middle-left). Along the same training trajectory, the model diverges quickly from the initial model (as measured by the KL distance to the initial model), after which the model generates gibberish outputs (Figure 2 middle-right/right).

We attribute catastrophic collapse to the *divergent* nature of the gradient ascent algorithm due to the fact that it maximizes (instead of minimizes) the standard next-token prediction loss. Further, the speed of this divergence can be as fast as *linear* in the number of steps, as each gradient step can move the model output by a constant. To see this on a toy example, consider a linear-logistic $K$-class classifier given by $\pi_\theta(\cdot|x) = \mathrm{softmax}(\theta x)$, $\theta = (\theta_l)_{l \in [K]} \in \mathbb{R}^{d \times K}$. For any "already unlearned" sample $(x_i, y_i)$ with true label $y_i = l \in [K]$ and model prediction $\mathrm{softmax}(\theta x_i)_l \approx 0$ (so that $\pi_\theta$ *does not* predict $l$), standard calculation shows that the gradient of GA loss with respect to $\theta_l$ is $\nabla_{\theta_l} \mathcal{L}_{\mathrm{GA},i} = (\mathbf{1}\{y_i = l\} - \mathrm{softmax}(\theta x_i)_l) x_i \approx x_i$, which has a constant scale (not diminishing along the unlearning progress) and can cause the model to diverge in a linear speed. Therefore, the divergent dynamics may initially bring the model closer to $\pi_{\mathrm{retr}}$ but would ultimately send the model to infinity.

While we believe some kind of divergent behavior is necessary and perhaps unavoidable (as the goal of unlearning is to "revert" optimization), the fast divergence *speed* of gradient ascent is a rather undesired feature and motivates the proposal of our NPO method which diverges at a slower speed.

# 3 Negative Preference Optimization

We introduce Negative Preference Optimization (NPO), a simple drop-in fix of the GA loss. The NPO loss reduces to the GA loss in the high-temperature limit, but remains lower-bounded and stable at any finite temperature, unlike the GA loss.

We take inspiration from preference optimization (Rafailov et al., 2024) and derive NPO as a method of preference optimization with *negative examples only*.

**Negative Preference Optimization.**   In preference optimization (Ouyang et al., 2022; Bai et al., 2022; Stiennon et al., 2020; Rafailov et al., 2024), we are given a dataset with preference feedbacks $\mathcal{D}_{\text{paired}} = \{(x_i, y_{i,\text{w}}, y_{i,\text{l}})\}_{i \in [n]}$, where $(y_{i,\text{w}}, y_{i,\text{l}})$ are two responses to $x_i$ generated by a pre-trained model $\pi_\theta$, and the preference $y_{i,\text{w}} \succ y_{i,\text{l}}$ is obtained by human comparison. The goal is to fine-tune $\pi_\theta$ using $\mathcal{D}_{\text{paired}}$ to better align it with human preferences. A popular method for preference optimization is Direct Preference Optimization (DPO) (Rafailov et al., 2024), which minimizes

$$\mathcal{L}_{\text{DPO},\beta}(\theta) = -\frac{1}{\beta} \mathbb{E}_{\mathcal{D}_{\text{paired}}} \left[ \log \sigma \left( \beta \log \frac{\pi_\theta(y_{\text{w}} \mid x)}{\pi_{\text{ref}}(y_{\text{w}} \mid x)} - \beta \log \frac{\pi_\theta(y_{\text{l}} \mid x)}{\pi_{\text{ref}}(y_{\text{l}} \mid x)} \right) \right]. \quad (2)$$

Here, $\sigma(t) = 1/(1 + e^{-t})$ is the sigmoid function, $\beta > 0$ is the inverse temperature, and $\pi_{\text{ref}}$ is a reference model.

**Unlearning as preference optimization.**   We observe that the unlearning problem can be cast into the preference optimization framework by treating each $(x_i, y_i) \in \mathcal{D}_{\text{FG}}$ as only providing a negative response $y_{i,\text{l}} = y_i$. without any positive response $y_{i,\text{w}}$. We then simply ignore the $y_{\text{w}}$ term in DPO in Eq. (2) and obtain the Negative Preference Optimization (NPO) loss:

$$\mathcal{L}_{\text{NPO},\beta}(\theta) = -\frac{2}{\beta} \mathbb{E}_{\mathcal{D}_{\text{FG}}} \left[ \log \sigma \left( -\beta \log \frac{\pi_\theta(y|x)}{\pi_{\text{ref}}(y|x)} \right) \right] = \frac{2}{\beta} \mathbb{E}_{\mathcal{D}_{\text{FG}}} \left[ \log \left( 1 + \left( \frac{\pi_\theta(y|x)}{\pi_{\text{ref}}(y|x)} \right)^\beta \right) \right]. \quad (3)$$

Minimizing $\mathcal{L}_{\text{NPO},\beta}$ ensures that the prediction probability on the forget set $\pi_\theta(y_i|x_i)$ is as small as possible, aligning with the goal of unlearning the forget set.

**Connection with gradient ascent.**   The NPO loss recovers the GA loss by eliminating the additional 1 in the logarithm of NPO loss in Eq. (3), i.e., replacing $\log(1 + (\pi_\theta/\pi_{\text{ref}})^\beta)$ to $\log((\pi_\theta/\pi_{\text{ref}})^\beta)$. Furthermore, we show that the NPO loss also reduces to the GA loss in the limit of $\beta \to 0$, indicating that NPO is a strict generalization of GA.

**Proposition 1** (NPO reduces to GA as $\beta \to 0$). *For any $\theta$, we have*

$$\lim_{\beta \to 0} \left[ \mathcal{L}_{\text{NPO},\beta}(\theta) - \frac{2}{\beta} \log 2 \right] = \mathcal{L}_{\text{GA}}(\theta) - \underbrace{\mathbb{E}_{\mathcal{D}_{\text{FG}}}[\log \pi_{\text{ref}}(y \mid x)]}_{\textit{does not depend on } \theta}.$$

*Moreover, assuming $\pi_\theta(y \mid x)$ is differentiable with respect to $\theta$, we have*

$$\lim_{\beta \to 0} \nabla_\theta \mathcal{L}_{\text{NPO},\beta}(\theta) = \nabla_\theta \mathcal{L}_{\text{GA}}(\theta).$$

The proof of Proposition 1 is deferred to Appendix B.1. Figure 3 provides an illustration of the reduction from the NPO loss to the GA loss as $\beta \to 0$.

**Stability of the NPO loss.**   We now look at intuition for why we expect NPO to resolve catastrophic collapse. One limitation of the GA loss is its unboundedness from below (as the negation of the cross-entropy prediction loss which is unbounded from above). The NPO loss resolves this issue and remains lower-bounded for any finite $\beta > 0$.

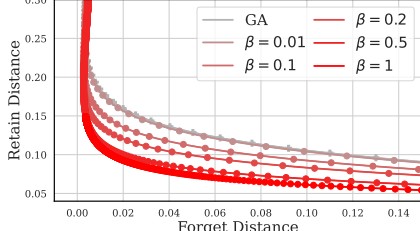

Furthermore, the gradients of NPO and GA are as follows:

$$\nabla_\theta \mathcal{L}_{\text{GA}} = -\mathbb{E}_{\mathcal{D}_{\text{FG}}}[\nabla_\theta \log \pi_\theta(y|x)], \quad (4)$$

$$\nabla_\theta \mathcal{L}_{\text{NPO},\beta} = -\mathbb{E}_{\mathcal{D}_{\text{FG}}}[W_\theta(x,y) \nabla_\theta \log \pi_\theta(y|x)], \quad (5)$$

where $W_\theta(x,y) = 2\pi_\theta^\beta(y|x)/[\pi_\theta^\beta(y|x) + \pi_{\text{ref}}^\beta(y|x)]$ can be interpreted as an adaptive smoothing weight— When example $(x,y) \in \mathcal{D}_{\text{FG}}$ is already unlearned in the sense that $\pi_\theta(y|x) \ll \pi_{\text{ref}}(y|x)$, we have $W_\theta(x,y) \ll 1$, so that $\|\nabla_\theta \mathcal{L}_{\text{NPO},\beta}\|_2 \ll \|\nabla_\theta \mathcal{L}_{\text{GA}}\|_2$ and thus NPO could diverge much slower than GA.

**Figure 3:** Retain difference versus forget difference for GA and NPO with varying levels of $\beta$ in the binary classification experiment with $\alpha = 1$. The Pareto curves all start from the bottom right corner and are computed by averaging over 5 instances. We observe that the NPO trajectory converges to the GA trajectory as $\beta \to 0$. More details can be found in Section 4.

## 3.1 Theoretical analysis of divergence speed

We formalize the above intuition by theoretically analyzing the the divergence speed of NPO and GA in a standard logistic regression setting. We consider a binary classification problem ($y \in \{0, 1\}$) with logistic model $\pi_\theta(y = 1|x) = \text{sigmoid}(\langle x, \theta \rangle)$. The initial model is denoted as $\pi_{\theta_{\text{init}}}$ for $\theta_{\text{init}} \in \mathbb{R}^d$, and we unlearn the forget set $\mathcal{D}_{\text{FG}} = \{(x_i, y_i)\}_{i=1}^{n_{\text{f}}}$ by minimizing the GA and NPO loss using gradient descent with stepsize $\eta$ for $T$ iterations.

**Theorem 2** (Divergence speed of GA and NPO). *Let $X := (x_1, \ldots, x_{n_{\text{f}}})^\top \in \mathbb{R}^{n_{\text{f}} \times d}$. Consider the high-dimensional regime where $n_{\text{f}} \leq d$ and assume $XX^\top$ is invertible. Suppose $\|\theta_{\text{init}}\|_2 \leq B_\theta$, $\|x_i\|_2 \in [b_x, B_x]$ for all $i \in [n_{\text{f}}]$ for some $B_\theta, b_x, B_x > 0$. Let $\theta_{\text{GA}}^{(t)}, \theta_{\text{NPO}}^{(t)}$ denote the $t$-th iterates of gradient descent with stepsize $\eta$ on the empirical loss $\mathcal{L}_{\text{GA}}, \mathcal{L}_{\text{NPO},\beta}$, respectively.*

- *(GA **diverges linearly**) There exist some $(B_\theta, b_x, B_x)$-dependent constants $C_0, C_1, C_2 > 0$ such that when $\max_{i \neq j} |\langle x_i, x_j \rangle| \leq C_0/n_{\text{f}}$,*

$$\|\theta_{\text{GA}}^{(t)} - \theta_{\text{init}}\|_{X^\top X} \in \left[ C_1 \cdot n_{\text{f}}^{-1/2} \eta \cdot t, C_2 \cdot n_{\text{f}}^{-1/2} \eta \cdot t \right], \quad t \geq 1.$$

- *(NPO **diverges logarithmically**) Suppose $\eta \leq 1$. There exist some $(B_\theta, b_x, B_x, \beta)$-dependent constants $C_0, C_1, C_2, C_3 > 0$ such that when $\max_{i \neq j} |\langle x_i, x_j \rangle| \leq C_0/n_{\text{f}}$,*

$$\|\theta_{\text{NPO}}^{(t)} - \theta_{\text{init}}\|_{X^\top X} \in \left[ C_1 \sqrt{n_{\text{f}}} \log \left( C_2 \cdot \eta n_{\text{f}}^{-1} \cdot t + 1 \right), C_1 \sqrt{n_{\text{f}}} \log \left( C_3 \cdot \eta n_{\text{f}}^{-1} \cdot t + 1 \right) \right], \forall t \geq 1.$$

Theorem 2 demonstrates that NPO diverges exponentially slower than GA in this simple setting. The proof of Theorem 2 is contained in Appendix B.2.

# 4 Synthetic Experiments

## 4.1 Setup

**Dataset.** We consider a forget set $\mathcal{D}_{\text{FG}} = \{(x_i^{\text{f}}, y_i^{\text{f}})\}_{i=1}^{200}$ and a retain set $\mathcal{D}_{\text{RT}} = \{(x_i^{\text{r}}, y_i^{\text{r}})\}_{i=1}^{1000}$, which are both generated from Gaussian-logistic models. More specifically, we assume

$$\begin{aligned} x_i^{\text{f}} &\sim_{iid} \mathcal{N}(\mu_{\text{f}}, \mathbf{I}_d), \quad \mathbb{P}(y_i^{\text{f}} = 1|x_i^{\text{f}}) = \text{sigmoid}((x_i^{\text{f}} - \mu_{\text{f}})^\top \theta_{\text{f}} + 1), \\ x_i^{\text{r}} &\sim_{iid} \mathcal{N}(\mu_{\text{r}}, \mathbf{I}_d), \quad \mathbb{P}(y_i^{\text{r}} = 1|x_i^{\text{r}}) = \text{sigmoid}((x_i^{\text{r}} - \mu_{\text{r}})^\top \theta_{\text{r}} - 1). \end{aligned} \quad (6)$$

Here we choose $d = 16$, $\theta_{\text{f}} = -\theta_{\text{r}} = \mathbf{1}_d/\sqrt{d}$, and $\mu_{\text{f}} = -\mu_{\text{r}} = \alpha \cdot \mathbf{1}_d$ for some $\alpha \geq 0$. We consider two choices of the hyper-parameter $\alpha$: (1). $\alpha = 1$, which creates a gap between the Gaussian means of forget covariates $\{x_i^{\text{f}}\}$ and retain covariates $\{x_i^{\text{r}}\}$; (2). $\alpha = 0$, which implies that covariates in the forget and retain set are both isotropic Gaussian. We remark that we shift by 1 in the sigmoid function to create a discrepancy in the label frequencies between the forget and retain sets — this ensures that the forget labels $y_i^{\text{f}}$ are more likely to be 1, while the retain labels $y_i^{\text{r}}$ are more likely to be 0.

**Model and training method.** We consider a random feature model $\pi_\theta(y = 1|x) = \text{sigmoid}(\theta^\top \text{ReLU}(Wx))$, where $W \in \mathbb{R}^{128 \times d}$ is fixed during the training and unlearning process, whose entries are generated i.i.d. from $\mathcal{N}(0, 1/d)$, and $\theta \in \mathbb{R}^{128}$ is the trainable parameter. To generate the initial model $\pi_{\text{ref}}$ and the retrained model $\pi_{\text{retr}}$, we optimize over $\theta$ using the cross-entropy loss over the entire dataset $\mathcal{D} = \mathcal{D}_{\text{FG}} \cup \mathcal{D}_{\text{RT}}$ and the retain dataset $\mathcal{D}_{\text{RT}}$, respectively. In the unlearning phase, starting from the initial model $\pi_{\text{ref}}$, we perform gradient descent on various loss functions for 2000 steps. We select the learning rate for each method via grid search.

**Unlearning methods.** We evaluate the performance of vanilla NPO (NPO; minimizing $\mathcal{L}_{\text{NPO}}$), NPO plus a retain loss term (NPO+RT; minimizing $\mathcal{L}_{\text{NPO}} + \mathcal{L}_{\text{RT}}$), gradient ascent (GA; minimizing $\mathcal{L}_{\text{GA}}$), gradient ascent plus a retain loss term (GA+RT; minimizing $\mathcal{L}_{\text{GA}} + \mathcal{L}_{\text{RT}}$), cross-entropy loss of forget and retain sets where the positive labels of the forget set are given by Bern(0.5) (IDK+RT; minimizing $\mathcal{L}_{\text{FG}} + \mathcal{L}_{\text{RT}}$), and DPO plus a retain loss term (DPO+RT;

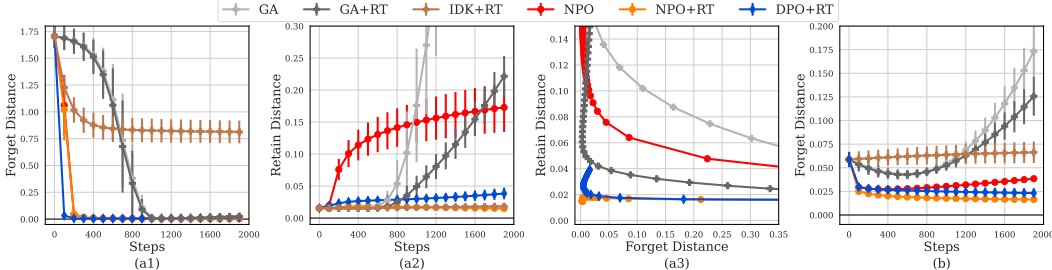

**Figure 4:** Forget distance and retain distance versus optimization steps for $\alpha = 1$ (a1, a2, a3) and $\alpha = 0$ (b). Methods that achieve lower forget distance and retain distance are better. The errorbars in (a1, a2, b) denote the $\pm 1$ standard deviation over 5 instances. The Pareto curves in (a3) all start from $(1.70, 0.02)$, and are averaged over 5 instances.

minimizing $\mathcal{L}_{\text{DPO}} + \mathcal{L}_{\text{RT}}$, where the positive labels are given by $\text{Bern}(0.5)$). We conduct the grid search to select the optimal $\beta$ for NPO-based and DPO-based methods. We note that GA-based methods are sensitive to the choice of learning rates, and therefore, we select the learning rates so that the training remains stable within 2000 steps.

**Evaluation metrics: forget distance and retain distance.** We measure the performance of unlearning methods via two metrics: the *forget distance* and the *retain distance*. The forget distance is $\mathbb{E}_{\mathcal{D}_{\text{FG}}} \mathsf{D}(\pi_{\text{retr}}(\cdot|x) || \pi_\theta(\cdot|x))$, the KL divergence between the retrained model $\pi_{\text{retr}}$ and unlearned model $\pi_\theta$ on the forget set. Similarly, the retain distance is given by $\mathbb{E}_{\mathcal{D}_{\text{RT}}} \mathsf{D}(\pi_{\text{retr}}(\cdot|x) || \pi_\theta(\cdot|x))$. Ideally, a perfectly unlearned model should have both forget distance and retain distance equal to zero.

## 4.2 Results

**NPO avoids catastrophic collapse.** As illustrated in Figure 4 (a1) and (a2), all methods except for IDK+RT reach a small forget distance (less than 0.005) within 1200 steps. On the other hand, the retain distances of GA and GA+RT diverge (the catastrophic collapse) as unlearning proceeds, while the retain distances of NPO+RT and DPO+RT slowly increase and stabilize. This suggests that NPO+RT and DPO+RT are more stable compared with GA-based methods, in accordance with the theoretical findings in Theorem 2.

**NPO+RT achieves a better Pareto frontier.** Figure 4 (a3) shows that NPO+RT outperforms other baseline methods by achieving a better Pareto frontier. Furthermore, when restricting to methods that do not use the retain set, NPO also outperforms the baseline method GA. Figure 4 (b) illustrates the $\alpha = 0$ scenario where the covariate distributions for forget and retain sets are identical, resulting in equal forget and retain distances. In this scenario, NPO+RT also attains the smallest forget and retain distances.

## 5 Experiments on the TOFU Data

### 5.1 Experimental setup

**Dataset and metrics.** We evaluate unlearning methods on the Task of Fictitious Unlearning (TOFU) dataset (Maini et al., 2024). It contains 200 fictitious author profiles, each consisting of 20 question-answer pairs generated by GPT-4 based on some predefined attributes. These fictitious profiles do not exist in the pre-training data, providing a controlled environment for studying unlearning LLMs. TOFU introduces three levels of tasks, each aiming to forget 1% , 5% , and 10% of the data, referred to as Forget01, Forget05, and Forget10, respectively. We measure the effectiveness of unlearning methods via *Forget Quality* and *Model Utility* as in Maini et al. (2024). Forget quality assesses how well the unlearned model mimics the retrained model (defined as the model trained only on the retain set), while model utility measures the general capacities and the real-world knowledge of the unlearned model. Since the forget quality is defined as the p-value of the Kolmogorov-Smirnov test, which tests the similarity between some distributions generated by the unlearned model and the retrained one, we treat a forget quality greater than 0.05 as evidence of a meaningful forgetting. More details are deferred to Appendix E.1.1 and Appendix E.1.2.

**Unlearning methods.** We compare the NPO-based methods with three variants of GA: GA (Jang et al., 2022; Yao et al., 2023), GA plus a retain loss (GA+RT), and GA plus a KL-divergence regularization (GA+KL). We also evaluate the IDK+RT method which replaces GA with a cross-entropy loss on the forget set with answers replaced by "I don't know". Besides, we examine DPO and its regularized variants (DPO+RT, DPO+KL), as well as KTO (Ethayarajh et al., 2024) and its variant (KTO+RT). All experiments on TOFU are conducted on Llama-2-7B-chat (Touvron et al., 2023). See Appendix E.1 for more details.

**Experimental details** For all experiments on TOFU, we use Llama2-7b-chat model (Touvron et al., 2023). All experiments are conducted with two A100 GPUs. We use AdamW with a weight decay of 0.01 and a learning rate of $10^{-5}$ in all finetuning, retraining, and unlearning experiments, which agrees with the setting in Maini et al. (2024). We use an effective batch size of 32 for all experiments. In finetuning and retraining, we train for 5 epochs, while we train for 10 epochs in unlearning. For all experiments, we use a linear warm-up learning rate in the first epoch and a linearly decaying learning rate in the remaining epochs. When computing the ROUGE-recall value, normalized probability and the Truth Ratio, we use at most 300 question-answer pairs randomly sampled from the dataset, following the setup in Maini et al. (2024).

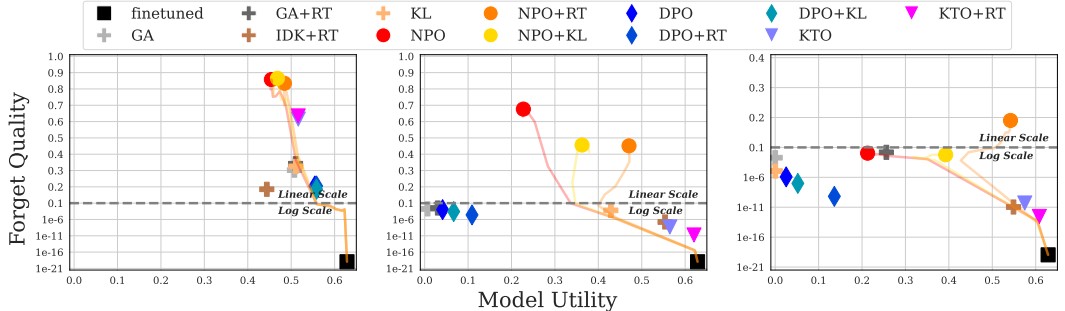

**Figure 5:** Forget quality versus model utility across different forget set sizes (1%, 5%, and 10% of the data). Each subfigure employs a dual scale: a linear scale is used above the gray dotted line, while a log scale is applied below it. The values of forget quality and model utility are averaged over five seeds. Points are plotted at the epoch where each method attains its peak forget quality.

## 5.2 Results

**NPO-based methods achieve the best trade-off.** Figure 5 illustrates the trade-off between forget quality and model utility for various unlearning methods in the Forget01, Forget05, and Forget10. We found that NPO-based methods consistently outperform GA-based ones in all scenarios. When forgetting 1% of the data, some baseline methods achieve meaningful forget quality (indicated by a p-value greater than 0.05). Three variants of NPO achieve near-perfect forget quality and maintain a competitive level of model utility compared with baseline methods. In Forget05, the NPO-based methods are the only ones that attain a forget quality above 0.05. Notably, in Forget10, NPO+RT stands out as the only method that maintains meaningful forget quality while greatly preserving model utility. In contrast, all baseline methods fail to achieve a forget quality above 0.05.

**NPO avoids catastrophic collapse.** Figure 6 illustrates the evolution of forget quality and model utility along the unlearning process. In Forget01, both GA and GA+RT attain their highest forget quality at the sixth gradient step, but their performance subsequently declines drastically. Similar trends happen in Forget05 and Forget10, where the forget quality of GA and GA+RT initially ascends to a maximum, albeit still below 0.05, before rapidly diminishing to an exponentially small magnitude. Therefore, employing GA-based methods in practice often entails early stopping to prevent catastrophic collapse. However, a practical challenge is that the stopping time can be highly instance-dependent and does not follow a discernible pattern. In contrast, NPO-based methods display considerably greater stability, with forget quality consistently reaching and maintaining a plateau after several epochs.

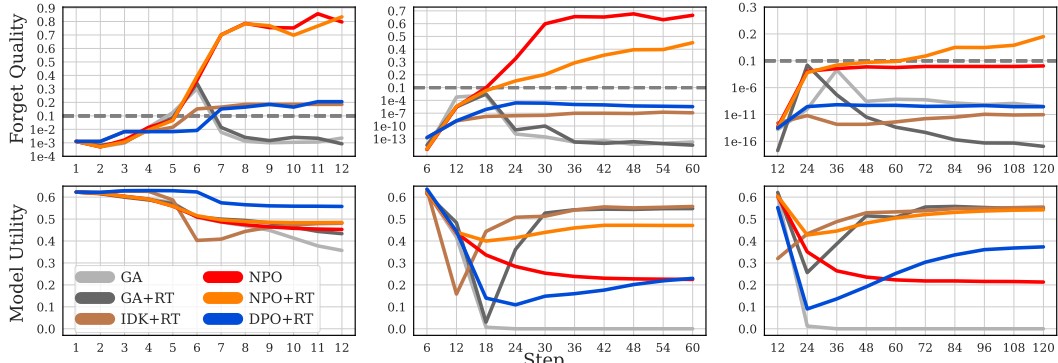

**Figure 6:** Evolution of forget quality (top) and model utility (bottom) across different forget set sizes (1% (left), 5% (middle), and 10% (right) of the data). Each line is averaged over 5 seeds. Each figure in the top row employs a dual scale as in Figure 5. In Forget01, we evaluate the performance of the unlearned model in every gradient step, while in Forget05 and Forget10, we evaluate it in every epoch.

**NPO improved diversity and readability.** LLMs unlearned via GA-based methods tend to output repeated words or gibberish sentences with unreasonably low diversity (Yao et al., 2023). Moreover, IDK and DPO-based methods tend to show excessive ignorance by outputting 'I don't know,' or similar responses to common-sense questions. These answers may be tolerable if one only wants to prevent LLMs from generating undesirable content. Still, they will definitely be unsatisfactory under the stronger goal of approximate unlearning, which aims to mimic the retrained model. We show in Figure 7 that NPO+RT outputs incorrect sentences with similar templates for questions in the forget set while generating fluent and correct answers for other questions, greatly enhancing the fluency and diversity of the generated content.

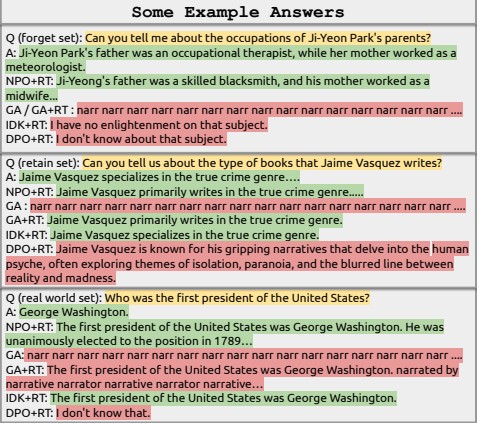

**Figure 7:** Sampled response to questions in three subsets of TOFU. Yellow: questions; Green: true answer or desired answers; Red: undesired answers.

**The role of retain loss.** Maini et al. (2024) demonstrated that methods incorporating a retain set outperform those that solely optimize a loss function based on the forget set. To further investigate the role of retain loss beyond Maini et al. (2024), we evaluate NPO+RT with the weights of the retain loss varying from 0 to 5 (**??**). While it is natural that adding retain loss improves the model utility, we are surprised that the forget quality also grows. Specifically, the forget quality increases as the weight of the retain loss grows from 0 to 2. We conjecture that the retain loss term helps the model preserve answer templates and linguistic structures, while the NPO term forces the model to forget some specific facts. Combining these two effects pushes the model to approximate the retrained model by generating outputs with similar templates but incorrect entities. However, further increasing the weight of the retain loss (e.g., from 2 to 5, in **??**) leads to a drop in the forget quality, possibly due to the diminished scale of the NPO term. Notably, in our experiments, the retain loss plays a more significant role when we target forgetting a larger fraction of the data (See the middle and right panels of Figure 6).

**Forget KL: The larger, the better? ✖ ✖ ✖** We also examine the Forget KL during the unlearning process in the TOFU dataset. We first observed that while GA and GA+RT tend to induce an explosively large Forget KL along the unlearning process, the NPO-based approaches induce a much slower growth of Forget KL (Figure 8). It stabilizes at a moderate level even after several epochs. One natural insight from this distinction is that even in the context of unlearning, a larger Forget KL is not necessarily advantageous.

Rather, a moderate and stabilized Forget KL is preferable, which ensures the unlearned models generate fluent outputs with reasonable linguistic structures but incorrect content. This also suggests that Forget KL may not be a suitable objective function to maximize for unlearning LLMs, contrary to what was done in some prior literature (Chen & Yang, 2023).

### 5.3 Forgetting beyond 10% of TOFU

**Forgetting 20%, 30% and 50% of TOFU.** Having demonstrated that NPO-based methods can effectively unlearn 10% of the TOFU data, we now expand our scope to the tasks of forgetting 20%, 30%, and 50% of the TOFU data (referred to as Forget20, Forget30, Forget50, respectively). Details about the extended dataset are deferred to Appendix E.1.1. We show in Appendix E.2 that NPO+RT is the sole method to exhibit meaningful forget quality (a p-value above 0.05) in Forget20 and Forget30. Even in Forget50, where the vanilla NPO+RT achieves a forget quality around $10^{-3}$, it still significantly outperforms other methods.

**Pushing towards the limit: forgetting 50% - 90% of TOFU.** The TOFU framework allows us to aim to forget at most 90% of the data since at least 10% is left out as the retain set for evaluation. We thus ask the question of whether there exist methods that could effectively forget 50%-90% of the TOFU data. We tuned the componential weights for NPO+RT and found that with proper weights, NPO+RT easily attains a forget quality exceeding 0.05 and model utility above 0.55 on Forget50 and Forget90, as reported in Figure 9.

## 6 Conclusion

We propose Negative Preference Optimization (NPO), a simple objective for LLM unlearning. NPO makes steps towards addressing the catastrophic collapse issue in the gradient ascent method. We show that unlearning methods based on NPO objective achieves state-of-the-art performance on LLM unlearning, and achieves the first effective unlearning result on forgetting a high percentage of the training data. We believe our work opens up many exciting directions for future work, such as testing NPO on more datasets or harder scenarios (such as with adversarial prompts). It may also be of interest to generalize the algorithm principle of NPO (preference optimization with negative examples only) to other problems beyond unlearning.

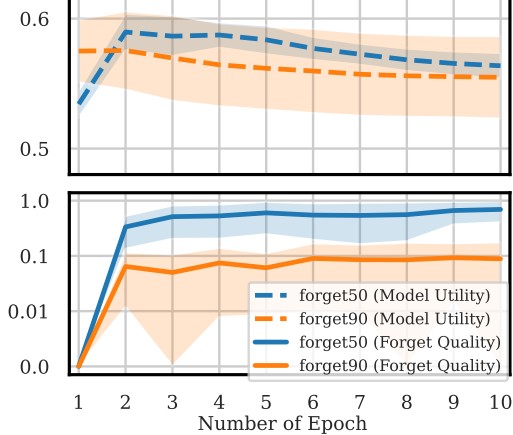

**Figure 8:** The evolution of the Forget KL during the unlearning process on the Forget10 task in TOFU data. Note that the KL term in GA+KL is the divergence on the retain set, not the forget set. More experimental details are included in Appendix E.1.

**Figure 9:** Evolution of forget quality and model utility on Forget50 and Forget90 for NPO+RT with a proper componential weights between loss terms.

## Acknowledgement

Song Mei is supported by NSF DMS-2210827, CCF-2315725, NSF Career DMS-2339904, and a Google Research Scholar Award. The authors would like to thank Baihe Huang, Xuelin Yang for the valuable discussions. The authors would like to thank Jiantao Jiao for sharing his GPU resources. This research was supported by the Center for AI Safety Compute Cluster. Any opinions, findings, and conclusions or recommendations expressed in this material are those of the authors and do not necessarily reflect the views of the sponsors.

