# OpenReview forum: "Negative Preference Optimization: From Catastrophic Collapse to Effective Unlearning"
_colmweb.org/COLM/2024/Conference — COLM_

### Official Review · Reviewer_8opk · 2024-05-10

**Rating:** 8
**Confidence:** 3
**Ethics Flag:** 1

**Summary:**

This paper presents negative preference optimization (NPO), which is inspired by Direct Preference Optimization (DPO) and uses the forget set as negative responses, and leaves the positive response empty. Authors did experiments on synthetic binary classification and TOFU unlearning tasks and find NPO achieves strong performance. Authors also theoretically prove progression toward catastrophic collapse is exponentially slower than the gradient ascent method.

**Questions To Authors:**

N/A

**Reasons To Accept:**

1. Simple, intuitive, and effective method, supported by both theoretical proof and empirical studies.
2. Strong unlearning performance while maintaining model utility.

**Reasons To Reject:**

1. Important task vector unlearning [1] baselines are missing.
[1] Ilharco, Gabriel, et al. "Editing models with task arithmetic." The Eleventh International Conference on Learning Representations. 2022.
[2] Zhang, Jinghan, Junteng Liu, and Junxian He. "Composing parameter-efficient modules with arithmetic operation." Advances in Neural Information Processing Systems 36 (2023): 12589-12610.

2. The motivation for synthetic experiments is not very clear to me, and the connection between synthetic experiments and TOFU experiments is also not very clear. Would be nice to clarify the motivation and why it is necessary in this work.

---

> ### Author Rebuttal · Authors · 2024-05-31
>
> We thank the reviewer for the positive assessment and the insightful feedback. Below, we aim to address the points raised in the review.
>
> >Important task vector unlearning [1] baselines are missing. [1] Ilharco, Gabriel, et al. "Editing models with task arithmetic." The Eleventh International Conference on Learning Representations. 2022. [2] Zhang, Jinghan, Junteng Liu, and Junxian He. "Composing parameter-efficient modules with arithmetic operation." Advances in Neural Information Processing Systems 36 (2023): 12589-12610.
>
> We will include the reference in the paper. However, we may not be able to compare NPO with these methods in this paper, as the two vector unlearning methods notably differ from gradient-type methods and are therefore challenging to implement in a short time frame. We are happy to compare the performance of these methods with NPO in future work.
>
>
> > The motivation for synthetic experiments is not very clear to me, and the connection between synthetic experiments and TOFU experiments is also not very clear. Would be nice to clarify the motivation and why it is necessary in this work.
>
> We believe our synthetic experiments complement the TOFU experiments and serve as a proof of concept, as several observations (e.g., NPO avoids catastrophic collapse and achieves a better tradeoff) in the TOFU experiments also appeared in the simple synthetic experiments. Moreover, these observations can be demonstrated in a more quantifiable way in the synthetic experiments. For example, we are able to easily and reasonably define the forget and retain distances based on the prediction accuracy on the forget and retain distributions in the synthetic experiments, while we have to resort to more heuristic and artificial metrics (e.g., the forget quality and model utility) in TOFU tasks. Additionally, our synthetic experiments are on a classification task instead of a text generation task, suggesting the potential applicability of NPO to general probability-based prediction tasks.

---

### Official Review · Reviewer_FPT9 · 2024-05-10

**Rating:** 7
**Confidence:** 4
**Ethics Flag:** 1

**Summary:**

The paper addresses the problem how to make the model forget after it has been trained on a dataset. Previous methods fail to effectively unlearn due to drastic degradation on the model's general capability/utilities.

Inspired by the DPO work, the paper introduces NPO which essentially is a variant of DPO without the positive example. Due to the contrast against a reference model, the training becomes more stable and the resulting model retrains better its original utilities. Synthetic and real experiments demonstrate the empirical effect of the method. Paper also provided rigid theoretical analysis.

Paper is well written, easy to read.

Theoretically speaking, the introduced method seems to be a slight modification of DPO. Empirically speaking, the method is simple and appears effective to address the problems (eg model utility degradation).

**Questions To Authors:**

none

**Reasons To Accept:**

- Paper is well written and easy to understand.
- Method is simple but empirically effective, with experiments support.
- Paper additionally provided rigid analysis and proof.

**Reasons To Reject:**

- Essentially, the introduced method seems to be a slight modification/specialization of DPO.

---

> ### Author Rebuttal · Authors · 2024-05-31
>
> We thank the reviewer for engaging with our work and their valuable feedback. We acknowledge that NPO resembles and is motivated by DPO. However, there is a key difference: the goal of DPO is alignment and it requires pairwise comparison responses, whereas the goal of NPO is unlearning and it requires only a single response to be forgotten.

---

### Official Review · Reviewer_yYsm · 2024-05-11

**Rating:** 6
**Confidence:** 4
**Ethics Flag:** 1

**Summary:**

This paper proposes Negative Preference Optimization (NPO), which applies Direct Preference Optimization (DPO) for unlearning certain concepts. It analyzes the differences between Gradient Ascent (GA) and NPO, and shows better performance than GA on the TOFU dataset.

**Questions To Authors:**

"data already unlearned" contributes to stability.
However, during the unlearning process, the randomness in training could lead to a larger gradient size for NPO. How does the size of the gradient relate to stability? Can a larger gradient size be considered unstable?
2. The validity of the TOFU dataset and the metric used in the main experiment: The paper proposing this dataset has not been peer-reviewed and is only an arXiv preprint. The main experiment seems to heavily rely on an arxiv preprint paper.  It is reasonable to utilize the experimental setting and dataset which was already published in the other conference or journal such as [2].
3. There exist no baseline methods in the main experiment. I understand that the main baseline is GA, but I think it is necessary to evaluate the performance of the methods mentioned in Section 2.
[2] Jang et al., Knowledge Unlearning for Mitigating Privacy Risks in Language Models, ACL 2023.

**Reasons To Accept:**

1. DPO was appropriately applied to unlearning.
2. Section 2 is comprehensively written and easy to understand.
3. The proposed NPO is a generalization of Gradient Ascent, and it presents evidence of superior performance from the perspective of divergence.
4. The proposed approach demonstrated superior performance than Gradient Ascent on the TOFU dataset.

**Reasons To Reject:**

1. The name of the proposed method seems to be misleading. Since NPO does not use y_w, it cannot be considered as preference tuning, which causes confusion.
2. Despite mentioning various previous studies in Section 2, only a few baselines are used as a baseline approach in the main experiment. For example, the experiment doesn’t include the performance of [1].
3. The models used in the analysis are simple linear layers. It is uncertain whether the same results would be obtained using the Transformer architecture, which is commonly used in LLMs.
4. The validity of the TOFU dataset and the metric used in the main experiment is questionable. The paper proposing this dataset has not been peer-reviewed and is an arXiv preprint.
[1] Chen et al., Unlearn What You Want to Forget: Efficient Unlearning for LLMs, EMNLP2023.

---

> ### Author Rebuttal · Authors · 2024-05-31
>
> We thank the reviewers for their helpful feedback and insightful comments. Below, we address these points.
>
> > The name of the proposed method seems to be misleading…
>
> We named NPO since it closely resembles and is motivated by DPO. Additionally, NPO can be viewed as a preference optimization algorithm that dis-favors the forget set (negative response), and thus (relatively) favoring responses dissimilar to the negative response. We will add more discussion in the paper to clarify the name.
>
> > Only a few baselines are used as a baseline approach in the main experiment…
>
> In Sec 5 and Appendix D, we tested several other methods besides GA, including combinations of GA, KL, retain loss, and non-GA-based methods such as DPO and KTO. After submission, we also tested EUL loss in [1]. In Forget01, at the end of the 10th epoch, EUL achieves a forget quality of 0.0003 while NPO+RT achieves 0.834, both averaging over 5 seeds. Intuitively, the GA components of [1] are problematic: the loss function is unbounded, causing gradient dynamics to diverge quickly. Moreover, there is a forget KL term in the EUL loss function. In Appendix C, we have shown it is better to keep a moderate forget KL than a very large one. This suggests that forget KL may not be an ideal objective for unlearning tasks. The results also show that the method in [2] does not perform as well as NPO variants in TOFU tasks. We are happy to include them in the final version.
>
> >The models used in the analysis are simple linear layers. It is uncertain whether the same results would be obtained using the Transformer architecture...
>
> The linear setting in our theoretical analysis and synthetic experiments serves as a proof of concept: several experimental observations in the TOFU also appear in this simplest problem setup, i.e., NPO avoids catastrophic collapse and achieves a better tradeoff. We believe extending the theoretical results to the transformer architecture is beyond the scope of this work but would be an interesting direction for future work.
>
> > The paper proposing this dataset has not been peer-reviewed and is only an arXiv preprint.
>
> While benchmarks for LLM unlearning (arguably) have not converged yet, we feel TOFU provides a clear definition of the unlearning goal (i.e., mimicking the output of the retrained model) and designs reasonable metrics to evaluate the model’s forget quality and retain utility. Therefore, we chose this dataset even though it is a recent work on arXiv.

---

> > ### Comment · Reviewer_yYsm · 2024-06-05
> > **Thank you for your response.**
> >
> > Thanks for the response.
> > It would be good to further describe how the size of the gradient relates to stability in the manuscript.
> > Since most of the questions have been addressed, I revised my evaluation as 6.

---

> > > ### Author Response · Authors · 2024-06-05
> > > **Relation between "stability" and "size of the gradient"**
> > >
> > > We greatly appreciate your raising your score!
> > >
> > > Are you asking about the relationship between our terminology of “stability” and “the size of the gradient”? We say an unlearning algorithm is stable if all samples are unlearned, the gradient should vanish. Otherwise, if the gradient is still large after all samples are unlearned, the parameters will diverge quickly. GA is unstable because the gradient remains non-vanishing even after a sample is effectively unlearned. NPO will be much more stable because, for unlearned samples, its gradient will be small, causing the model to stop training as soon as all samples are unlearned.

---

### Official Review · Reviewer_v7L5 · 2024-05-12

**Rating:** 7
**Confidence:** 2
**Ethics Flag:** 1

**Summary:**

This paper presents an application of direct policy optimization for unlearning knowledge from a large language model (LLM). Experimental results on TOFU demonstrate that the proposed method improves over existing state-of-the-art methods.

**Reasons To Accept:**

- The paper studies an interesting and important problem.
- The paper is well-written.
- The experimental findings are interesting.

**Reasons To Reject:**

- Analysis over a wider range of datasets in different domains would be very beneficial, as well as conducting experiments with different model architectures

---

> ### Author Rebuttal · Authors · 2024-05-31
>
> We thank the reviewer for engaging with our work and their valuable feedback. We agree that evaluating our method on various datasets and models is beneficial. Following the submission, we assessed the performance of NPO on the Phi model, which was also evaluated in the TOFU paper. The experimental results align with that of Llama.
>
> More specifically, due to the limited time in the rebuttal phase, we tried the tasks of Forget01 and Forget10 on the Phi model. The fine-tuning and unlearning process follows what was described in the TOFU paper. For each task, we unlearn the model for 10 epochs using a learning rate of 2*10^-5 and a linear warmup in the first epoch.
>
> In Forget10, we observe a very similar trend as in Llama2-7B model. GA achieves an extremely low Forget Quality along the unlearning process, with the maximal forget quality around 1.83 * 10^-7. GA+RT performs better than GA, but shows a very unstable unlearning trajectory: the forget quality arises somewhere in the middle and then drops drastically. At the end of the 10th epoch, it attains a forget quality of 5.5*10^-9. On the contrary, NPO+RT shows the best forget quality throughout the whole process. In the meantime, the forget quality steadily grows when we unlearn it for more epoch. For example, at the end of the 5-th epoch, the forget quality for NPO+RT is 0.055, and at the end of the 10-th epoch, it is around 0.147, which is a significant number (above 0.05). Meanwhile, the model utility of NPO+RT is high and ranks the highest two among all unlearning methods.
>
> In Forget01,  we observe a similar phenomenon. The Forget Quality for NPO and NPO+RT keeps growing along the whole process and finally exceeds 0.5, while the forget quality for GA initially grows but falls quickly and finally achieves a value lower than 0.05.
> Due to the space limit and the limitation that we cannot modify the paper or attach a link during the rebuttal phase, we are not able to show you the figure for the evolution of forget quality and model utility. But we are happy to include these additional experimental results in the final version of our paper.

---

> > ### Comment · Reviewer_v7L5 · 2024-06-06
> >
> > Thank you for your response and for providing additional results. I will keep my score as it is.

---

### Decision · Program_Chairs · 2024-07-10

**Decision:**

Accept

**Comment:**

This work introduces negative preference optimization to unlearn a target dataset. Experimental results show that the proposed methods outperformed existing baselines.

As reviewers pointed out, this work studies a very interesting problem with theoretical results. The paper is also well-written especially Section 2. The method also demonstrates strong unlearning performance while maintaining model utility.

Reviewers also suggest a few revision comments such as the motivation around synthetic experiments, as well as applicability of the current method on other model architectures.

The authors have provided detailed clarification and responses to some of these during the discussion period, and I’d like to encourage the authors to further address these in their revised version.